# Lost in the Middle: An Emergent Property from Information Retrieval Demands in LLMs

## Abstract

The performance of Large Language Models (LLMs) often degrades when crucial information is in the middle of a long context, a "lost-in-the-middle" phenomenon that mirrors the primacy and recency effects in human memory. We propose that this behavior is not simply a flaw indicative of information loss but an adaptation to different information retrieval demands during pre-training: some tasks require uniform recall across the entire input (a long-term memory demand), while others prioritize the most recent information (a short-term memory demand). Consistent with this view, we show that this U-shaped performance curve emerges when LLMs (GPT-2 and Llama variants) are trained from scratch on two simple human memory paradigms simulating long-term and short-term memory demands. Our analysis reveals that while the recency effect directly aligns with short-term memory demand in the training data, the primacy effect is induced by the uniform long-term memory demand and is additionally influenced by the model's autoregressive properties and the formation of attention sinks. Our main findings from simple human memory paradigms also generalize to a sequence completion task, which more closely resembles the next-token prediction process in LLM pre-training. Together, our findings reveal how information retrieval demands, model architecture, and structural attention dynamics during model training can jointly produce positional bias observed in LLMs.

## 1 Introduction

When answering questions over exceedingly long context information, Large Language Models (LLMs) exhibit a "lost-in-the-middle" phenomenon in which accuracy drops significantly for information near the center of the context window (Liu et al., 2023). This phenomenon is strikingly similar to serial position effects found in human memory literature (Figure 1), where people preferentially recall items from the *beginning (primacy)* and *end (recency)* of a study list with higher accuracy, producing a characteristic U-shaped curve (Murdock & Bennet, 1962). Despite the lost-in-the-middle effect being reproduced and studied in a variety of contexts and tasks (Janik, 2023; Hsieh et al., 2024a), a complete understanding of its underlying mechanisms has yet to be established, with evidence pointing to the role of LLMs' intrinsic attention biases (Hsieh et al., 2024b; Xiao et al., 2023; Gu et al., 2024) and architectural biases (Wu et al., 2025). While much of the work on the lost-in-the-middle effect has considered it a model bias and focused on eliminating the effect altogether (Hsieh et al., 2024b; Zhang et al., 2024; Wang et al., 2024), our current work provides an alternative perspective, considering it as an emergent property under the information retrieval demands during LLM pre-training.

An LLM's ability to perform real-world tasks using its context window critically depends on retrieving the correct contextual information in the first place (Veseli et al., 2025). While the role of information retrieval demands during LLM pre-training and its connection to lost-in-the-middle behavior remains unclear, cognitive psychology offers a vast literature to understand human behavior under different memory demands. This literature primarily distinguishes between the short-term memory demand, when a task requires recalling recent events (Bunting et al., 2006), and the long-term memory demand, when a task requires recalling events further in the past (Murdock & Bennet, 1962; Roberts, 1972). Theoretical frameworks such as rational analysis (Anderson, 1990) and resource-rational analysis (Lieder & Griffiths, 2020) are used to understand if specific behaviors are emergent properties that arise from meeting task demands under cognitive architectural constraints. From this

perspective, many cognitive behaviors once considered biases or flaws are now understood as rational adaptations to environmental challenges (Lieder et al., 2018; Callaway et al., 2024; Huttenlocher et al., 2000). Similarly, an LLM's behavior is shaped by the interplay between its model architecture and the goal it was trained to accomplish (McCoy et al., 2024).

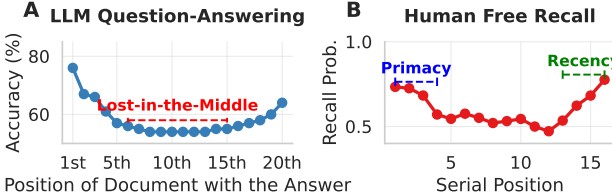

Figure 1: (A) The "lost-in-the-middle" behavior in LLMs, where accuracy drops significantly for information near the center of the context window. (B) Serial position effects in human memory, where items from the beginning (primacy) and end (recency) of a study list are recalled with higher accuracy, producing a characteristic U-shaped curve.

Within this framework, the recency effect, as observed in the human memory literature, has been interpreted as a rational adaptation to the short-term memory demand in the environment, where recent information is more important and more likely to reappear (Anderson & Milson, 1989). This hypothesis is supported by observations that the forgetting curve in human memory aligns with statistical patterns found in real-world environments like news articles, emails, and social media posts (Anderson et al., 2022; Anderson & Milson, 1989). In contrast, when memory demands are placed uniformly across an entire sequence, theoretical analysis shows that the primacy effect, emphasizing recall from the beginning of a sequence, emerges as an optimal strategy for maximizing memory performance (Zhang et al., 2021). Together, primacy and recency effects contribute to the serial position effects, or lost-in-the-middle behavior, commonly observed in human memory. They are not cognitive flaws, but adaptive behaviors that support task performance.

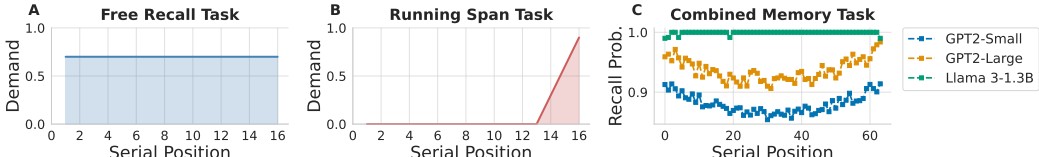

Figure 2: Lost-in-the-middle behavior in LLMs arises from adaptations to short-term and long-term memory demands during training. **(A)** The *free recall* task involves recalling all items from the presented sequence in any order, which places a *long-term memory demand* equally across the entire list. **(B)** The *running span* task involves recalling the last $N$ items preceding a specified location (i.e., recall token), which places a *short-term memory demand* on only the most recent information. **(C)** Our findings reveal that when LLMs are trained jointly on both tasks from scratch, lost-in-the-middle behavior emerges.

Inspired by the human memory literature, our research examines whether the lost-in-the-middle behavior in LLMs arises from similar principles: a rational adaptation to short-term and long-term information retrieval demands under architectural constraints. Supporting this hypothesis, we show that lost-in-the-middle behavior emerges when LLMs (GPT-2 and Llama-3.2 variants in our work) are *trained from scratch* on two classic human memory tasks (Figure 2C). We use the free recall task (i.e., recalling a sequence in any order; Figure 2A) to induce long-term information retrieval demand and the running span task (i.e., recalling only the last few items of a sequence in any order; Figure 2B) to induce short-term information retrieval demand. Although other combinations of tasks and data distributions may also give rise to the lost-in-the-middle behavior after model training, here, we present a minimal set of task demands where the lost-in-the-middle behavior emerges purely from task optimization. To further validate our findings, we replicated our results using a masked sequence completion task, which more closely resembles the next-token prediction process in LLM pre-training. We use two different variations of this task to replicate the long-term and short-term memory demands imposed by the memory tasks: one where the masked subsequence can come from

anywhere in the original sequence (long-term information retrieval demand) and one where masked subsequences only appear near the end of the list (short-term information retrieval demand).

While the recency effect (higher end-of-list recall in Figure 2C) aligns with the shape of short-term information retrieval demand in the training data (Figure 2B), it is less intuitive why the primacy effect (higher beginning-of-list recall in Figure 2C) emerges from the long-term information retrieval demand placed uniformly across an entire sequence (Figure 2A). We hypothesize that the primacy effect arises from the interaction between the uniform long-term retrieval demand and the autoregressive nature of LLMs, specifically the causal masking that biases attention toward earlier tokens. Past work has linked positional bias observed in LLMs with causal masking (Wu et al., 2025). If the primacy effect arises from the combination of a uniform long-term retrieval demand and the autoregressive nature of LLMs enabled by causal masking, then we should expect the same training process to produce this effect in other autoregressive architectures. Consistent with our hypothesis, we found that the primacy effect emerges when a uniform, long-term retrieval demand is paired with an autoregressive architecture (RNNs), but not with a bidirectional encoder-decoder (T5), suggesting that both the task demand and causal-style processing are necessary conditions for primacy.

In addition to architectural biases, we hypothesize that attention sinks are a key mechanism linking transformer attention dynamics to the lost-in-the-middle behavior. Attention sinks describe the phenomenon where the initial tokens of a sequence disproportionately attract most of the attention weight across several attention heads, despite carrying little semantic content (Xiao et al., 2023). They appear throughout the training process across a broad range of architectures, model scales, and tasks, suggesting they are byproducts of fundamental elements of the transformer architecture (Gu et al., 2024). Given the previously established links between attention sinks and positional bias in transformers, we conducted an ablation study in which we disrupted attention sinks throughout models trained on each of the memory tasks. Although attention sinks emerge consistently across all our tasks, disrupting them had selective effects: it eliminated the primacy effect and impaired performance on the free recall task (long-term memory demand), but had no impact on the running span task (short-term memory demand). These results indicate that attention sinks are an important mechanism for supporting tasks that place long-term memory demands.

To summarize our contributions, we identified a minimal set of task demands, long-term memory demand, and short-term memory demand, that produce lost-in-the-middle behavior. We trained GPT-2 (Small/Large) and Llama-3.2 1B from scratch on two classic memory paradigms simulating these task demands, and reproduced primacy under the free recall task, recency under the running span task, and U-shape behavior when the two tasks are trained jointly.

## 2 METHODS

### 2.1 TASK DEFINITIONS

To investigate the effects of different information retrieval demands, we train GPT-2 Small, GPT-2 Large, and Llama-3.2 1B on three memory tasks: Free Recall, Running Span, and Combined Free Recall and Running Span (i.e., jointly training Free Recall and Running Span), as well as a masked sequence completion task (full formal definitions can be found in the Appendix A.1). Each task presents a list of discrete items, $W_{\text{presentation}} = (w_1, ..., w_M)$, between sequence tokens <SoS> and <EoS>, and differs only in what the model is asked to retrieve.

**Free Recall (FR).** After the list presentation, the model is expected to output all items from the list in any order. That is, for a presented sequence of the form $I_{\text{FR}} = \begin{bmatrix} \texttt{<SoS>} & W_{\text{presentation}} & \texttt{<EoS>} \end{bmatrix}$, the expected response is any unordered set of the original items in the list. This imposes a uniform long-term information retrieval demand across the list (Fig. 2A).

**Running Span (RS).** The presented list of items is followed by a cue token <RECALL_n>, with the model input taking the form: $I_{\text{RS}} = \begin{bmatrix} \texttt{<SoS>} & W_{\text{presentation}} & \texttt{<RECALL\_n>} & \texttt{<EoS>} \end{bmatrix}$. Based on the cue token found in the sequence, the model is expected to output the last $n$ items that precede the cue, in any order. In our experiments, each trial has a value of $n$ randomly sampled between 1 and 7, with items nearer to the cue token being included in relatively more trials than items farther away. This concentrates short-term demand near the end of the list (Fig. 2B).

**Combined (FR+RS).** In this task, the presented sequence is equivalent to that of the running span task, but with the model expected to perform two separate recall tasks. The model is expected to (i) recall the last $n$ items (order-agnostic) and (ii) recall the entire list (order-agnostic). This mixes uniform long-term memory demand with an end-weighted short-term memory demand, yielding a mixed demand condition. During combined-task training, the free-recall and running-span objectives were optimized jointly using equal loss weighting within sample batches.

**Masked Sequence Completion.** For the masked sequence completion task, after presenting the list, we reveal a contiguous subsequence from the study list followed by blanks, with model input taking the following form: $I_{\text{SCT}} = \begin{bmatrix} \texttt{<SoS>} & W_{\text{presentation}} & \texttt{<EoS>} & w_s, \ldots, w_{s+r-1}, & \underbrace{\_ \cdots \_}_{b \text{ blanks}} \end{bmatrix}$. Based on this presented sequence, the model is expected to fill the blanks with the next $b$ items in original order as they are presented in the list. We test three sampling regimes to mirror memory demands imposed by the three memory tasks: (i) Uniform (positions chosen uniformly), (ii) Recency-weighted (later positions sampled more often), and (iii) Combined (one uniform prompt and one recency-weighted prompt per trial). Full details of how this sampling is performed can be found in the Appendix.

## 2.2 IMPLEMENTATION AND BEHAVIORAL MEASURES

We train GPT-2 Small, GPT-2 Large, and Llama-3.2-1B on each of the described memory tasks, using randomly shuffled target sequences to encourage order-agnostic recall. In order to assess the effect of architectural bias on "lost-in-the-middle" behavior, we train and evaluate an RNN-based seq2seq and T5 encoder-decoder model on the free recall task. For all tasks, we use sequence lengths of 64 items, i.e., randomly sampled nouns in the memory tasks and randomly sampled single symbols (e.g., '#', 'G', '9', etc.) in the masked sequence completion tasks, and train all models from random initializations on 100,000 randomly sampled sequences for 25 epochs. For the memory tasks (not including the masked sequence completion task), we introduce 10 random shuffles of each target recall sequence during model training.

To evaluate the model behavior elicited by each task, we apply analytical tools from cognitive psychology traditionally used to study human memory: serial position curves, probability of first recall, and conditional response probability (Murdock & Bennet, 1962; Kahana, 1996).

**Serial position curves (SPC)** tracks recall accuracy as a function of item position in the input list, typically revealing primacy and recency effects. Formally, the probability that an item from serial position $i$ in the study list is recalled at all during the recall period is given by $P_{\text{SPC}}(i) = \frac{1}{N} \sum_{n=1}^{N} R_{n,i}$, where $N$ is the number of trials, and $i$ is the serial position in the list, where $i \in \{1, 2, \ldots, L\}$. The indicator variable $R_{n,i}$ is equal to 1 if the item at position $i$ in trial $n$ is recalled, and 0 otherwise.

**Probability of first recall (PFR)** measures where in the list recall tends to begin, offering insights into the model's initial output strategy. The probability that the first item recalled comes from serial position $i$ is given by $P_{\text{PFR}}(i) = \frac{1}{N} \sum_{n=1}^{N} F_{n,i}$, where $F_{n,i}$ is an indicator variable that equals 1 if, in trial $n$, the first recalled item was presented at position $i$, and 0 otherwise.

**Conditional response probability (CRP)** characterizes the patterns of recall transitions. Formally, CRP at lag $t$ is the probability that, after recalling an item at position $i$ in the list, the next recalled item comes from position $i + t$. This is computed as the number of observed transitions with lag $t$ divided by the number of possible transitions with lag $t$, i.e., $CRP(t) = \frac{\text{observed}_t}{\text{possible}_t}$. The numerator counts all actual recall transitions with lag $t$, while the denominator corresponds to opportunities where the item at position $i + t$ had not been recalled yet. For example, in a list $W = (w_1, \ldots, w_5)$ with corresponding recalled sequence $(w_3, w_1, w_4)$, the transition $w_3 \rightarrow w_1$ contributes to a lag of $-2$ and $w_1 \rightarrow w_4$ contributes to lag $+3$. For a lag of $+1$, no transitions occur, but there is one possible opportunity ($w_3 \rightarrow w_4$) resulting in $CRP(+1) = \frac{0}{1} = 0.0$.

# 3 RESULTS

## 3.1 LOST-IN-THE-MIDDLE ARISES FROM JOINT OPTIMIZATION ON SHORT-TERM AND LONG-TERM MEMORY DEMANDS

In this section, we examine whether the lost-in-the-middle behavior in LLMs can emerge from optimal adaptation to tasks with different information retrieval demands. Figure 3 shows the behavioral results when training each model on three memory tasks: the free recall task (long-term memory demand), the running span task (short-term memory demand), and the joint training of free recall and running span tasks (mixed memory demand).

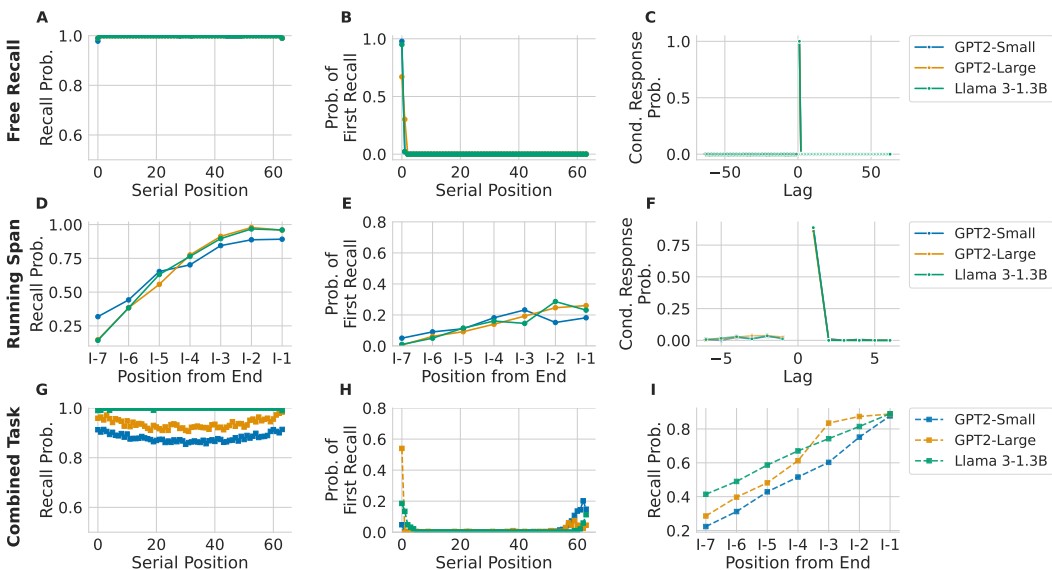

Figure 3: Recall behavior results for all models across each task experiment. (A-C) Serial position curve, probability of first recall, and conditional response probability for each model on the free recall task. (D-F) Relative-to-end recall probability (i.e., recall probability for positions offset from the <RECALL_n> token), probability of first recall, and conditional response probability for each model on the running span task. (G-I) Serial position curve (free recall response), probability of first recall (free recall response), and relative-to-end recall probability (running span response) when models are trained simultaneously on the free recall and the running span tasks.

When trained from scratch on the free recall task, all models displayed near-perfect recall performance (Figure 3A). Their behavior mimicked the classic human primacy effect, characterized by a strong tendency to initiate recall from the beginning of the list (Murdock & Bennet, 1962, Figure 3B), and a tendency to recall items in consecutive order (Kahana, 1996, Figure 3C). In contrast, models trained on the running span task demonstrated recency effects (Figure 3DE), specifically, higher recall probabilities for items relatively closer to the end of the list (Murdock & Bennet, 1962), indicating a short-term information retrieval demand.

The most intriguing recall patterns emerge under the combined training regime. For GPT-2 models, the serial position curve shifts toward a U-shape, exhibiting both primacy and recency effects, which in turn resulted in a lost-in-the-middle behavior (Figure 3G). Though Llama-3.2 1B continues to perform nearly flawlessly on the overall recall performance (Figure 3G), its probability of first recall indicates that it initiates recall from both the beginning and the end of the list (Figure 3H), suggesting a change to its underlying recall behavior similar to that of the smaller GPT-2 models.

These results align with a growing body of work showing that lost-in-the-middle weakens with increasing model scale (Guo & Vosoughi, 2024; Liu et al., 2023). Larger models distribute attention more evenly and generate more uniform recall accuracy, which reduce the visibility of primacy and recency in serial position curves. These findings support our hypothesis that the lost-in-the-middle behavior can emerge from optimal adaptation to short-term and long-term information retrieval demands during model training.

## 3.2 PRIMACY RELATES TO ARCHITECTURAL BIASES

While the recency effect aligns well with the shape of short-term information retrieval demand in the training data, it is less obvious why the primacy effect emerges from the long-term information retrieval demand placed uniformly across an entire list. To test whether the primacy effect – which emerges from optimizing models on a free recall task (Figure 3B) – is additionally shaped by causal masking in LLMs, we train two additional models on the same task: an autoregressive recurrent seq2seq model and a bidirectional T5 encoder–decoder. The autoregressive RNN-based seq2seq model exhibits strong primacy effects with near-perfect recall near the beginning of the list (Figure 4A), and a high probability of initiating recall from the first item of the sequence (Figure 4B). It also demonstrated a preference for transitioning forward through the sequence, as evidenced by the high conditional response probability for +1 lags (Figure 4C). In contrast, T5 lacks the primacy effect, with about equal probability of initiating recall from anywhere in the sequence (Figure 4DE). The behavioral differences between these two models suggests that the primacy effects seen in decoder-only LLMs and RNNs may largely stem from their autoregressive design, while models like T5, without this constraint, avoid such biases.

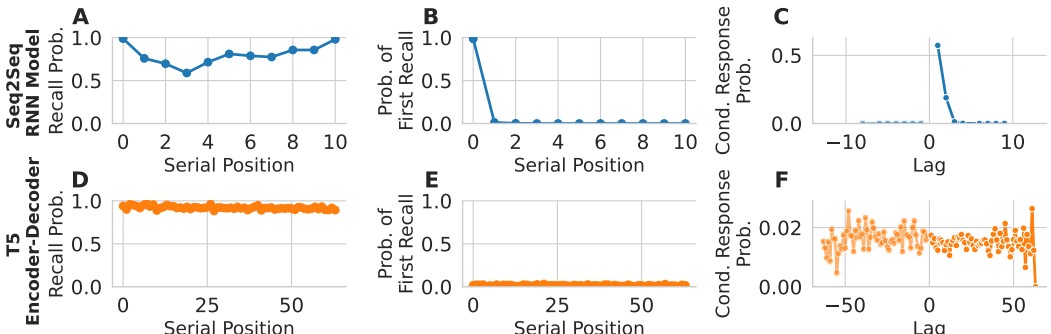

Figure 4: Free recall behavior for alternative model architectures. (A-C) Free recall behavior for an RNN-based seq2seq model. This is an example of another autoregressive model that exhibits the primacy effect similar to decoder-only LLMs. (D-F) Free recall behavior for T5. This encoder-decoder model exhibits a flat recall curve and a uniform probability of first recall.

## 3.3 LINKING PRIMACY BEHAVIOR TO ATTENTION SINKS

Although we have established that alternative autoregressive models exhibit similar primacy biases, the underlying cause for this bias in decoder-only transformers, such as GPT-2, is not immediately apparent. By disproportionately focusing on the beginning of the sequence, attention sinks may be a possible mechanism for anchoring recall to early tokens. If so, ablating these sinks should weaken primacy while leaving recency-focused performance relatively unaffected. We examined the potential functional role of attention sinks in our memory tasks by adopting a quantitative metric from (Gu et al., 2024), which proposed a threshold-based method for identifying and measuring attention sinks across transformer layers and heads. For each attention head $h$ in layer $l$, the importance score for the $k$-th token is defined as the average attention it receives across all tokens from position $k$ to the end of the sequence of length $T$:

$$\alpha_h^l(k) = \frac{1}{T - k + 1} \sum_{i=k}^{T} A_{i,k}^l \tag{1}$$

An attention head is considered to exhibit an attention sink if $\alpha_h^l(k)$ exceeds a chosen threshold, $\epsilon$. Using this metric, we analyzed each model and task condition in our experiments. Figure 5A-C presents heatmaps of attention weights for heads deemed attention sinks at various sink metric values. To understand the functional role that attention sinks may play in the positional bias observed in LLMs, we conducted a set of intervention experiments. We performed targeted disruptions by applying dropout to entire attention layers identified as exhibiting attention sink behavior. Layers were selected based on exceeding the attention sink threshold of $\varepsilon = 0.8$ on the first token, corresponding to the heatmap visualization in Figure 5C, which demonstrates clear attention sink behavior. We

chose this threshold because only $\varepsilon = 0.8$ cleanly isolates heads that exhibit characteristic attention sink behavior, whereas lower thresholds drastically increase the number of heads included in the ablation, leading to broad, nonspecific degradation of performance. Figure 5D-F depicts recall behavior results before and after the attention dropout, applied to the free recall, running span, and combined tasks. In the free recall task, the largest negative effect on performance was observed at the first token in all instances, consistent with the role of attention sinks in supporting primacy; additionally, the decline in performance extended across the entire sequence (Figure 5D). Our additional analyses (Appendix A.4) show that this negative impact on the entirety of the sequence is unique to attention sink dropout: disrupting attention at other positions throughout the sequence leads to only a local negative impact on recall performance, but only disrupting the first token (i.e., the attention sink) leads to negative performance across the entire sequence.

When we applied the same intervention to models performing the running span task (Figure 5E), we observed a much smaller impact on recall accuracy across all models, which were tested to be non-significant (Figure 5G). On the combined free recall and running span task (Figure 5F), we see both a significant drop in recall performance as well as a marked change in recall behavior across all models. Although the Llama model exhibits a reduction in performance only near the beginning of the list, similarly to the free recall task, the GPT-2 Small and Large models additionally see a complete loss of the U-shape in their recall curves. Not only do both models exhibit a significant drop in recall near the beginning of the list, but they also show a negative impact on recall performance across the entire list. Overall, we show that attention sinks removal selectively influences the performance of tasks with long-term information retrieval demands (the free recall task and the combined task) but not tasks with short-term information retrieval demands (the running span task), as shown in Figure 5G, and that removing attention sinks also removes the primacy effect. These findings provide a link between the lost-in-the-middle behavior and the underlying attention mechanisms.

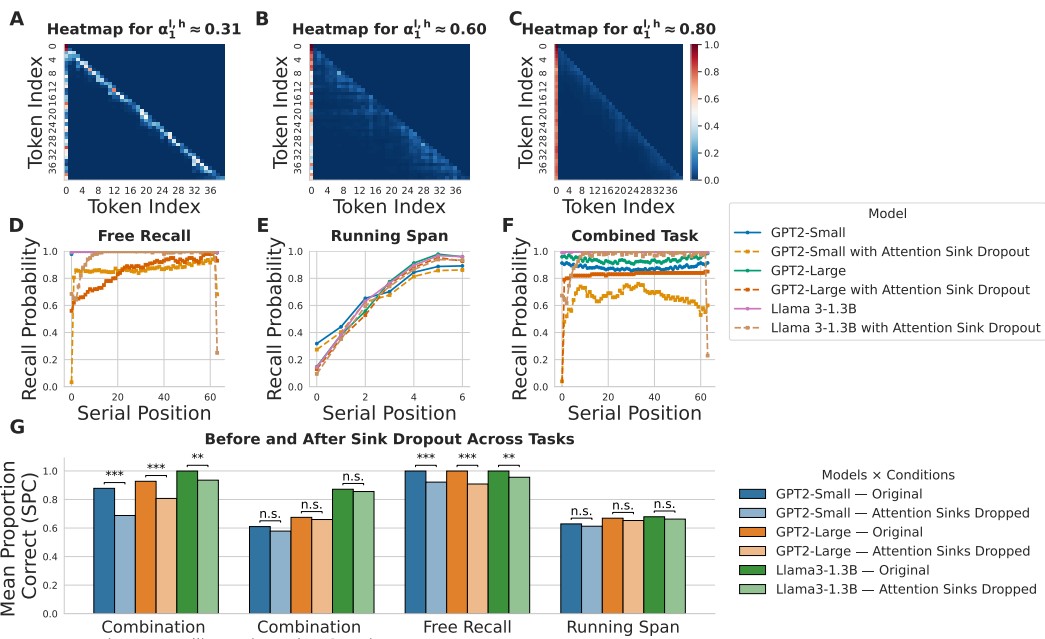

Figure 5: Attention sink and head ablation behavioral results. (A-C) These attention heatmaps show attention scores for sample heads identified as sinks at various thresholds. At $\epsilon = 0.8$, we see a clear attention sink form and use this threshold for ablation testing. (D-F) Recall behavior curves for each model on each task before and after attention sink head dropout. Both free recall and combined tasks show significant drops in performance, both at the primacy region and across the entire list. (G) Each bar represents the averaged recall accuracy of a model on a given task with or without attention sink dropout. For each pair of model-testing conditions, we perform a paired t-test (for aligned inputs) to determine the significance of the performance difference in the unablated and ablated performance metrics (* : $p < 0.05$, ** : $p < 0.01$, *** : $p < 0.001$, n.s. : not significant).

### 3.4 MASKED SEQUENCE COMPLETION TASK EXHIBITS SIMILAR POSITIONAL BIASES AS MEMORY TASKS

In the masked sequence completion task, we investigate whether the emergence of lost-in-the-middle behavior we observed in human memory paradigms can be generalized to a task that more closely resembles the next-token prediction process in LLM pre-training. If the same information retrieval demands and architectural biases are involved, we should expect to observe primacy, recency, and U-shaped recall patterns, along with effects of attention sink ablation. Importantly, by manipulating the position from which the target answer is drawn (uniform sampling, recency sampling, and a combination of uniform sampling and recency sampling), we can systematically impose memory demands analogous to those in the free recall and running span tasks. We analyze the models' accuracy and behavior as a function of the masked subsequence's position in the original sequence using the same behavioral metrics from our memory experiments. Results for all three task variations are shown in Figure 6A-C.

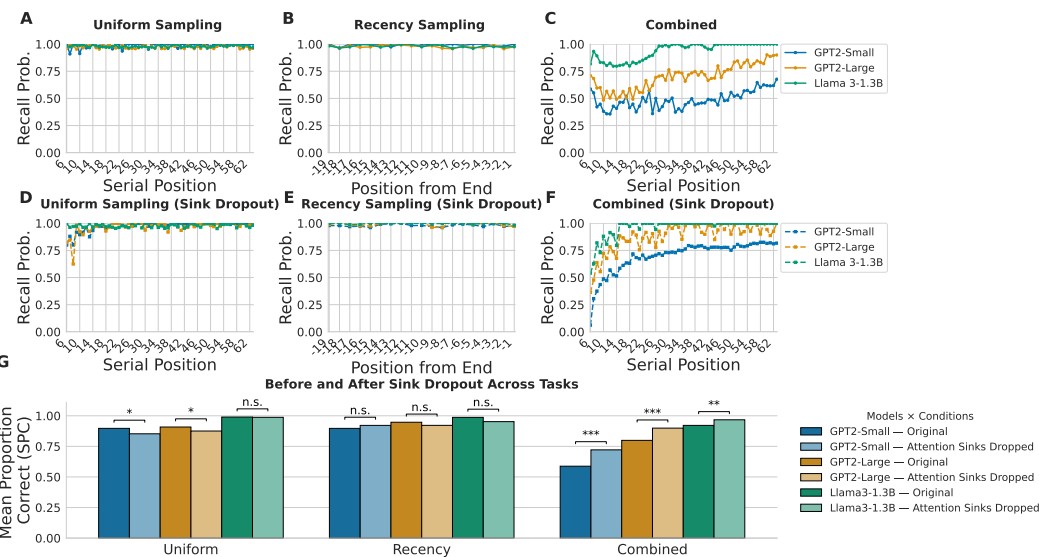

Figure 6: Model behavior and attention sink ablation results for three variants of the masked sequence completion task, simulating long-term information retrieval demand (uniform sampling), short-term information retrieval demand (recency sampling), and mixed information retrieval demand (combined sampling), respectively. (A-C) Serial position curves for each model across each of the three sampling conditions. (D-F) Serial position curves for each model across three sampling conditions with attention sink dropout, using a threshold value of $\epsilon = 0.8$. (G) Averaged model accuracy before and after attention sink dropout (* : $p < 0.05$, ** : $p < 0.01$, *** : $p < 0.001$, n.s. : not significant).

For all models, we see performance saturation in both the uniform- and recency-sampled conditions (Figures 6A-B), and additionally see the emergence of a characteristic U-shaped recall curve in the combined masked sequence completion task (Figure 6C). While both the GPT2-Small and Large models show a pronounced lost-in-the-middle behavior, the Llama-3.2 model exhibits a much smaller U-shaped curve, consistent with our previous observations in the memory experiments.

We repeat the attention sink dropout analysis for the masked sequence completion experiments, and evaluate each model on the corresponding tasks with attention heads ablated using the attention sink threshold of $\epsilon = 0.8$. The behavior results for models evaluated with attention head ablation are shown in Figures 6D-F, while the averaged performance results and significance tests are displayed in Figure 6G. Although not as pronounced as in the free recall experiment, we see a significant drop in performance in the uniformly-sampled sequence completion task for both GPT2-Small and Large, where both models show a drop in recall near the beginning of the list (depicted in Figure 6D). However, we do not see any significant drop in performance for the larger Llama-3.2 model, which is consistent with the negligible impact observed in the free recall task (Figure 5D). In the recency-sampled task (Figure 6E), no models show any significant change in recall performance or behavior, supporting the hypothesis that short-term memory demand tasks do not exhibit reliance on

attention sinks. Conversely, the combined sampling condition shows a significant effect of attention sink dropout on both performance (Figure 6G) and overall behavior (Figure 6F). Overall, we find that the model recall behaviors in three variants of sequence completion tasks align with the three memory tasks, with the combined training condition exhibiting the lost-in-the-middle behavior, and only the conditions with long-term information retrieval demands (uniform sampling and combined sampling) being significantly impacted by attention sink removal.

## 4 DISCUSSION

**Short-term and long-term memory demands explain lost-in-the-middle behavior**.  Our core finding is that lost-in-the-middle behavior can be induced in LLMs by manipulating their training objectives. Training models from scratch on a free recall task (uniform long-term memory demand) yields primacy, training on a running span task (end-weighted short-term memory demand) yields recency, and joint training on both tasks produces the canonical U-shaped curve associated with the lost-in-the-middle behavior (Liu et al., 2023). The fact that these effects emerge in simple task paradigms, without pre-training or confounding elements of natural text, strengthens the interpretation that they are consequences of optimization under task constraints rather than artifacts of specific datasets. This aligns with resource-rational perspectives in cognitive psychology (Lieder & Griffiths, 2020), which explain the emergence of primacy and recency effects as rational adaptations to environmental goals and computational constraints (Anderson & Milson, 1989; Zhang et al., 2021). Our serial position curves and the probability of first recall patterns closely mirror human data (Murdock & Bennet, 1962), pointing to future avenues in uncovering the connections between artificial and biological systems.

**Architectural biases shape serial-position curves**. We observe strong primacy in autoregressive models (RNN seq2seq and GPT-2), while a bidirectional encoder–decoder (T5) exhibits a flatter serial position curve and equal preference for initiating recall from anywhere in the sequence. These results agree with prior studies suggesting that autoregressive processing encourages concentrating more attention towards early tokens (Xiao et al., 2023; Wu et al., 2025), and that encoder–decoders trained on fixed-length sequences exhibit reduced positional biases (Liu et al., 2023). Model complexity also matters: we find that larger models (e.g., Llama-3.2 1B) exhibit reduced or eliminated U-shaped curves and maintain high overall recall, consistent with prior results that increased model complexity reduces lost-in-the-middle severity (Guo & Vosoughi, 2024; Liu et al., 2023). Together, these observations suggest that architectural biases and model scale interact with a task's information retrieval demand to produce the observed positional bias in LLMs.

Our findings also connect to recent mechanistic accounts of positional bias in transformers. Barbero et al. (2024) show that decoder-only models experience information over-squashing at long context lengths, where representations become increasingly insensitive to mid-sequence tokens. Barbero et al. (2025) further argues that strong attention to the first token arises as a stabilizing mechanism that mitigates this representational collapse, leading to the characteristic attention-sink pattern. Complementing these works, Wu et al. (2025) provide a graph-theoretic analysis demonstrating that causal masking and multi-layer attention amplify the influence of early tokens, even before incorporating positional encodings. They show that relative positional schemes such as RoPE partially counteract but do not eliminate this bias towards early position tokens, revealing architectural pressure toward primacy that is intrinsic in decoder-only transformers.

These mechanistic perspectives describe *why* early-token anchoring and positional asymmetries emerge from architectural constraints. Our results build on this foundation by identifying *when* these structural tendencies affect downstream task performance and behavior. Specifically, although attention sinks and over-squashing appear across tasks and model scales, we find that they only influence recall behavior under uniform long-term retrieval demands, and are largely irrelevant when the objective emphasizes more recent information. This helps reconcile why positional biases are visible in some settings and attenuated in others, even within the same architecture.

**Model complexity and attenuation of positional bias**

Our results help reconcile a counterintuitive result found in the literature: larger models show weaker, or no, U-shaped recall curves (Liu et al., 2023; Guo & Vosoughi, 2024), yet positional biases such as attention sinks and early-token anchoring remain detectable (Xiao et al., 2023; Gu et al.,

2024). In our experiments, Llama-3.2-1B demonstrates this pattern clearly. Although its serial recall accuracy is nearly flat, its probability of first recall still shifts under different retrieval demands in a manner similar to smaller models, and sink ablations selectively impair tasks with long-term retrieval requirements. Additional scale sweeps (Appendix A.2) extend this trend to Gemma-2 2B, Qwen-2.5 1.5B, and Llama-3.2 3B, with Llama-3.2 3B also showing shifted first recall under saturated recall performance, confirming that the U-shape recall pattern reduces consistently with increasing model size. We quantify this trend of decreasing lost-in-the-middle effect using a U-shape index capturing the gap between end- and mid-list recall performance. The index declines steadily from GPT-2 to Gemma-2, Qwen-2.5, and Llama-3.2, directly demonstrating that the U-shape pattern decreases with increasing model complexity (Appendix Fig. 8).

These findings suggest that scale mitigates the behavioral consequences of positional bias rather than eliminating the underlying mechanisms. This interpretation also aligns with modern long-context models (e.g., Gemini 1.5 Pro, Claude 3.5, and Llama 3) that achieve near-perfect recall across long sequences: increased capacity, improved positional encodings, and more diverse training distributions help compensate for the same architectural tendencies that produce primacy in smaller models.

**Attention sinks support primacy under long-term memory demand**. Attention sinks appear widely across transformers, but whether sinks are functionally meaningful remains debated. Some work argues they are largely dormant (Sandoval-Segura et al., 2025), others that they stabilize computation or can be harnessed for streaming or calibration along large context windows (Guo et al., 2024; Xiao et al., 2023; Yu et al., 2024). Using the thresholded sink metric adapted from Gu et al. (2024), our targeted ablations reveal a selective, functional contribution: disrupting attention sinks impairs tasks with long-term memory demands (free recall and the combined tasks), while leaving the short-term running span performance largely intact (running span task). The asymmetry in performance indicates that attention sinks play a direct role in the retrieval of information over the entire sequence. In contexts where the task demand is placed on more recent information, the system is comparatively insensitive to sink ablation, suggesting at least partially separable mechanisms for short-term versus long-term information retrieval in LLMs.

## 5 FUTURE WORK

Our study leaves a number of further evaluations for future work. A natural next step is to test the retrieval-demand framework directly on natural long-context benchmarks, such as re-ordering, context-insertion, and multi-document QA suites, to assess when the mechanisms we isolate predict improvements from existing positional-attention mitigations. Expanding the scaling analysis beyond the range we were able to include here would also clarify how primacy, recency, and attention-sink strength evolve across larger model families, in addition to how the underlying attention distribution may change with model scale. Our controlled experiments extend only to moderate sequence lengths, and prior work suggests that both positional anchoring and over-squashing intensify with increasing context window size. Systematically testing models at larger context window sizes would reveal whether the sink–primacy relationship strengthens monotonically, plateaus, or undergoes qualitative shifts at extreme lengths.

An important future direction is to evaluate how our retrieval-demand framework predicts when long-context mitigation strategies are effective. Prior work has introduced a variety of interventions aimed at reducing lost-in-the-middle behavior, including rotary-embedding rescaling (Zhang et al., 2024), attention-offsetting (Hsieh et al., 2024b), context reordering (Peysakhovich & Lerer, 2023), and positional-agnostic or modified attention mechanisms (Wang et al., 2024). Our results suggest that such methods may have the greatest impact on tasks dominated by uniform long-term retrieval demands, where primacy effects and attention sinks play a functional role. Evaluating these mitigation techniques across natural long-context tasks in the context of retrieval demands is another promising future direction.

It would also be valuable to investigate factors we did not sweep in this work, including alternative tokenization schemes, positional encodings (e.g., RoPE variants, ALiBi), and additional architectures such as state-space and hybrid models. Training under more realistic temporal or distributional regimes, such as recency-skewed data resembling web or news data, may help determine how natural statistics shape the balance of primacy and recency predicted by our framework.

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

# A APPENDIX

## A.1 FORMAL TASK DEFINITIONS

### A.1.1 FREE RECALL

A list of items, $W_{\text{presentation}}$, is presented between sequence tokens `<SoS>` and `<EoS>`. After the initial presentation, the model must output all presented items, in any order (order-agnostic recall). The task imposes memory demands uniformly across the entire sequence, as depicted in Figure 2A. We can formally define this task as follows:

Let $X \in \mathbb{R}^{T \times F}$ be a sequence of words , with start/end markers at indices $t_{\text{SoS}}$ and $t_{\text{EoS}}$, where $t_{\text{SoS}} < t_{\text{EoS}}$. Here, $T$ refers to the total length, in tokens, of the input sequence where $t_i$ refers to a particular token at position $i$, while $F$ is the embedding dimension of each input token. Inside the range $[t_{\text{SoS}} + 1, t_{\text{EoS}} - 1]$ lie $M \in \mathbb{N}_+$ item tokens $W = (w_1, \ldots, w_M)$, with each $w_i \in \{1, \ldots, F\}$. The target for this task is the multiset $\mathcal{W}_{\text{presentation}} = \{w_1, \ldots, w_M\}$, i.e. any unordered set of the original items appearing in the presentation list.

The form of each trial is as follows:

$$I_{\text{FR}} = \begin{bmatrix} \texttt{<SoS>} & \{w_1, \ldots, w_M\} & \texttt{<EoS>} \end{bmatrix}$$

### A.1.2 RUNNING SPAN

In this task, a list of items is presented with start/end tokens, defined similarly as in the free recall task, and an additional terminal cue token `<RECALL_n>`. The model is tasked with recalling the last $n$ items preceding this cue token, in any order. For our experiments, the value of $n$ is randomly sampled between 1 and 7 for each individual trial. As such, a recall token of $n = 3$ would have a ground-truth response of $w_{n-3}\ w_{n-2}\ w_{n-1}$ (with any order of these elements being acceptable), where $w_{n-x}$ corresponds to the word appearing $x$ positions before the recall token in the presented list. This sampling process will naturally lead to items closer to the recall token more frequently appearing in task trials, leading to the asymmetric memory demand curve appearing Figure 2B.

The task is defined formally as follows: Let $X \in \mathbb{R}^{T \times F}$ contain sequence tokens `<SoS>` at $t_{\text{SoS}}$, `<EoS>` at $t_{\text{EoS}}$, and a special recall cue token `<RECALL_n>` at $t_c$ with $t_{\text{SoS}} < t_c < t_{\text{EoS}}$. In our experiments, we only cue end-of-list recalls, such that $t_c = t_{\text{EoS}} - 1$. Items appear as a sequence $W = (w_1, \ldots, w_M)$ in $(t_{\text{SoS}}, t_{\text{EoS}})$. Each trial is presented in the following form:

$$I_{\text{RS}} = \begin{bmatrix} \texttt{<SoS>} & w_1, \ldots, w_M & \texttt{<RECALL_n>} & \texttt{<EoS>} \end{bmatrix}$$

Define $m_c = |\{i \in \{1, \ldots, M\} : \text{pos}(w_i) < t_c\}|$ and assume $n \leq m_c$. The target for the task is the multiset of possible sets of the target items

$$\mathcal{W}_n^{\text{pre}} = \{w_{m_c - n + 1}, \ldots, w_{m_c}\}.$$

A model must output any permutation of $\mathcal{W}_n^{\text{pre}}$, i.e., recall the tokens preceding the recall cue token in any order.

### A.1.3 COMBINED RUNNING-SPAN + FREE-RECALL

In the combined task condition, the cue `<RECALL_n>` appears at the end of the list in addition to standard start/end tokens, as previously described in the running span task. The model must (i) recall the last $n$ items that precede the cue (order-agnostic), and (ii) recall all items that appear in the entire list (also order-agnostic). This combined task condition imposes mixed memory demands, which include a uniform demand on all tokens (words) with an asymmetric increase to demand placed on the final 7 items of the list (as imposed by the running span portion of the task).

The formal definition is as follows: Let $X \in \mathbb{R}^{T \times F}$ contain `<SoS>` at $t_{\text{SoS}}$, `<RECALL_n>` at $t_c$, and `<EoS>` at $t_{\text{EoS}}$, with $t_{\text{SoS}} < t_c < t_{\text{EoS}}$. Items $W = (w_1, \ldots, w_M)$ lie between `<SoS>` and `<EoS>`. Each trial is presented in a form identical to the running span task:

$$I_{\text{COMBO}} = \begin{bmatrix} \texttt{<SoS>} & W_{\text{presentation}} & \texttt{<RECALL_n>} & \texttt{<EoS>} \end{bmatrix},$$

Let $m_c$ be the count of items before the recall cue and assume $n \leq m_c$. Define

$$\mathcal{W}_n^{\mathrm{pre}} = \{ w_{m_c-n+1}, \ldots, w_{m_c} \}, \qquad \mathcal{W}^{\mathrm{post}} = \{ w_1, \ldots, w_M \}$$

The target is the ordered pair of multisets $\left( \mathcal{W}_n^{\mathrm{pre}}, \mathcal{W}^{\mathrm{post}} \right)$. A model must output both multisets (order within each is irrelevant).

### A.1.4 MASKED SEQUENCE COMPLETION TASK

We draw inspiration from masked language modeling objectives widely used in pre-training, such as the masked sequence prediction task introduced in BERT (Devlin et al., 2019) and the span corruption objective in T5 (Raffel et al., 2019). In our adaptation, a list of items (individual symbols, in this case) is presented between <SoS> and <EoS>, after which a cue consisting of several items from the original list followed by blanks _ is shown. The formal definition of the task is as follows:

Let $X \in \mathbb{R}^{T \times F}$ be the sequence of the symbols with markers at $t_{\mathrm{SoS}} < t_{\mathrm{EoS}}$, and let the items within be $W = (w_1, \ldots, w_M)$. Choose integers $r \in \mathbb{N}_+$ (the revealed length of the sequence), $b \in \mathbb{N}_+$ (number of blanks), and a start index $s \in \{1, \ldots, M - r - b + 1\}$. The cue after <EoS> reveals the contiguous subsequence $(w_s, \ldots, w_{s+r-1})$ and then provides $b$ blanks. The target completion is the ordered tuple $C = (w_{s+r}, \ldots, w_{s+r+b-1})$, i.e. the $b$ items that follow the revealed items in the original sequence $W_{\mathrm{presentation}}$. The model must output the expected $b$ items in the order in which they were originally presented. The input format of this task can be written as:

$$I_{\mathrm{SCT}} = \begin{bmatrix} \texttt{<SoS>} & W_{\mathrm{presentation}} & \texttt{<EoS>} & w_s, \ldots, w_{s+r-1}, \underbrace{\_ \cdots \_}_{b \text{ blanks}} \end{bmatrix},$$

We present this task in three variations: uniform sampling, recency-weighted sampling, and combined sampling. In the uniform sampling condition, each cue window is chosen with equal probability, so that all items in the list are equally likely to be tested. This mirrors the uniform memory demand of the free recall task. In the recency-weighted sampling condition, cue windows are chosen with probability proportional to the recency of their blank positions. Formally, we can define a recency range $K \in \mathbb{N}_+$ (in our experiments $K = 7$) and a minimum sampling weight $\epsilon$. Each item position, $i \in \{1, \ldots, M\}$, is given a weight according to:

$$u(i) = \begin{cases} \epsilon, & i \leq M - K, \\ \epsilon + \dfrac{i - (M - K)}{K}, & i > M - K, \end{cases}$$

where this weight increases linearly toward the end of the list. For a cue window starting at index $s$ with $r$ revealed items and $b$ blanks, the window weight is defined as:

$$W(s) = \sum_{j=0}^{b-1} u(s + r + j)$$

which results in a sampling probability of:

$$\mathrm{Pr}(s) = \frac{W(s)}{\sum_{s'} W(s')}$$

This concentrates sampling on items nearer to the end of the list, matching the memory demand imposed by the running span task. In the combined sampling condition, each trial contains both a uniformly sampled cue window and a recency-weighted cue window, ensuring that all items are tested while ensuring the last $K$ items are sampled at a higher rate. This combined condition mirrors the demands imposed by the combined free recall and running span task.

## A.2 Additional Model Scale Experiments

To assess how positional effects evolve with model capacity, we trained several larger open-weight models on the same task suite used in the main paper: Gemma-2 2B, Qwen-2.5 1.5B, and Llama-3.2 3B. These models span distinct architectural design choices and positional-encoding variants, which allows us to examine whether the attenuation of the U-shape is primarily attributable to scale rather than to a specific architecture. For these experiments, we increased the list length from 64 items to 256 items to test whether longer contexts make the lost-in-the-middle effect observable in larger models. The results of these additional experiments are depicted in Figure 7.

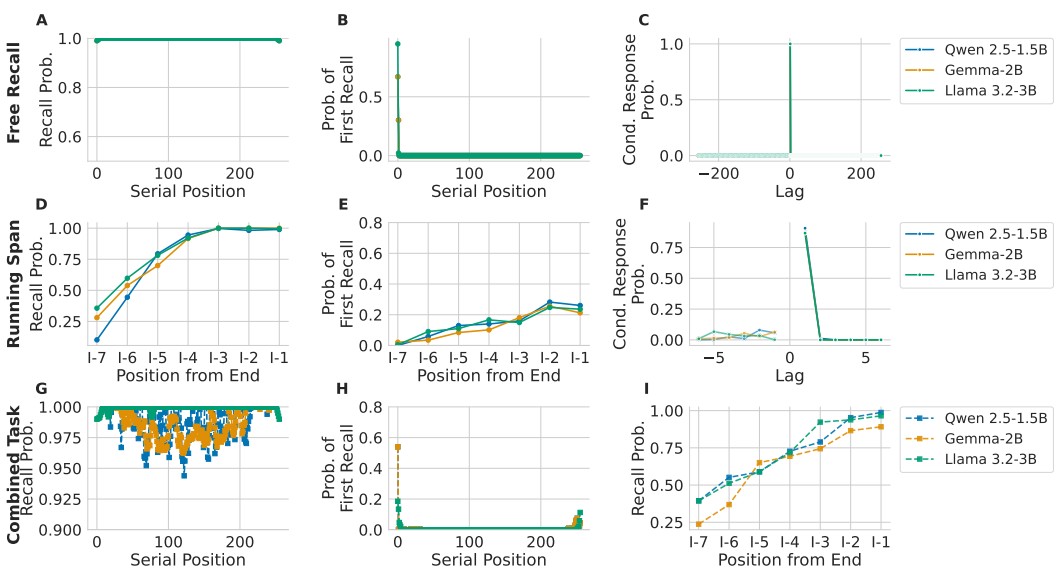

Figure 7: Recall behavior results for larger scale models across each task experiment. (A-C) Serial position curve, probability of first recall, and conditional response probability for each model on the free recall task. (D-F) Relative-to-end recall probability (i.e., recall probability for positions offset from the <RECALL_n> token), probability of first recall, and conditional response probability for each model on the running span task. (G-I) Serial position curve (free recall response), probability of first recall (free recall response), and relative-to-end recall probability (running span response) when models are trained simultaneously on the free recall and the running span tasks.

Across tasks, these models reproduced the qualitative patterns observed in smaller architectures but with substantially reduced magnitude. Free-recall accuracy was near-ceiling for all models, showing only weak primacy in Gemma-2 2B. Running span continued to elicit clear recency patterns, indicating that short-term retrieval demands remain relevant at larger model scales. The combined task revealed the sharpest distinctions: Gemma-2 2B and Qwen-2.5 1.5B showed a faint U-shape, while Llama-3.2 3B exhibited no visible U-shape. However, Llama-3.2 3B exhibits a shift in its first-recall distribution, indicated by an increase to first recall probability at the end of the list in the combined task case, that mirrors the strategy change observed in Llama-3.2 1B despite saturated recall performance.

In order to quantify the reduction in the lost-in-the-middle effect, we use a U-shape index heuristic computed as the mean recall at the first and last sequence positions minus the mean recall across the middle third of positions, shown in Eqn. 2 below.

$$U = \frac{R_1 + R_L}{2} - \frac{1}{|M|} \sum_{i \in M} R_i \qquad (2)$$

Where $R_i$ corresponds to the serial recall probability at position $i$, $L$ represents the list length, and $M$ is the set of positions in the middle-third of the sequence. Plotting this index against model size shows a smooth decline from GPT-2 through Gemma-2 and Qwen-2.5 to Llama-3.2 3B, confirming that positional bias weakens steadily with scale (Figure 8). Together, these experiments show that

the primacy–recency trade-offs identified in the main paper generalize to larger and more modern architectures, even though their increased capacity suppresses the overt U-shaped behavior.

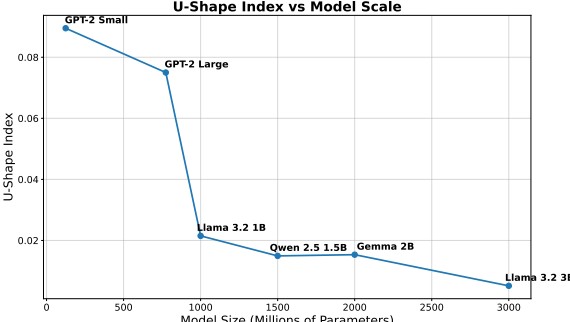

Figure 8: Comparison of model parameter count with the U-shape index indicates that models exhibit relatively less of a lost-in-the-middle effect as model complexity increases.

### A.3 ALTERNATIVE ATTENTION SINK ABLATION EXPERIMENTS

To test the robustness of the attention-sink findings and address concerns that layer-level dropout may remove computation unrelated to the sink mechanism, we conducted experiments with alternative ablation strategies. Results for these additional experiments are depicted in Figure 9.

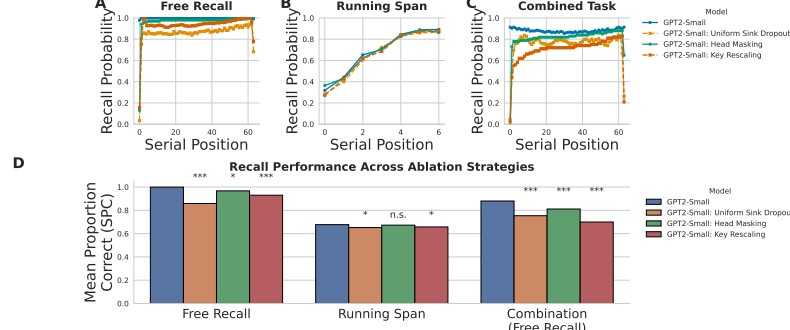

Figure 9: Attention sink and head ablation behavioral results with alternative methods and additional sink thresholds. (A-C) Recall behavior curves for each model on each task before and after attention sink head dropout with uniform sink layer dropout (as presented in the main paper), head masking, and key rescaling. While each ablation method leads to similar behavior shifts in performance, uniform layer dropout and key rescaling lead to larger impacts on performance than head masking. (D) Each bar represents the averaged recall accuracy of a model on a given task with or without attention sink dropout across the three methods analyzed. For each pair of model-testing conditions, we perform a paired t-test (for aligned inputs) to determine the significance of the performance difference in the unablated and ablated performance metrics (* : $p < 0.05$, ** : $p < 0.01$, *** : $p < 0.001$, n.s. : not significant).

We apply two alternative attention sink disruption methods in these additional experiments: head masking and key-positional rescaling. In head masking, for each identified sink head $h$, we replaced its attention matrix $A^{(h)}$ with 0 at inference time while preserving all other heads and residual-path computation. In key–positional rescaling, we modified only the keys of sink heads by applying a position-dependent scaling matrix $S$ whose entries reduce attention to only the first position, replacing $K^{(h)}$ with $SK^{(h)}$. Both approaches reproduced the central pattern observed with layer dropout: free-recall and combined-task performance declined primarily at early-list positions, while running-span accuracy showed only slight fluctuations from base performance. Key–positional rescaling produced the strongest effects and head masking the weakest, but all methods agreed on the selective impact of attention sink disruption. These results show that the behavioral effects attributed

to attention sinks are not artifacts of coarse ablations and persist under finer-grained manipulations targeted specifically on the attention sink.

### A.4 ATTENTION DROPOUT ACROSS SERIAL POSITIONS

In addition to attention sink dropout at token position 0, we also performed a series of trial evaluations for the long-term retrieval demand tasks (i.e., free recall in Figure 10A and uniformly-sampled sequence completion in Figure 10B) with attention disrupted at various positions throughout the sequence. We find that disrupting attention at specific positions in the sequence leads to a drop in recall performance at the position corresponding to the disrupted attention, as well as the positions immediately before and after the disrupted position. However, only when attention is disrupted on the first token of the sequence (i.e., the attention sink) do we see a negative impact on recall that extends across the entirety of the input sequence. This disparity in the disruption effect provides evidence that the attention sink has a role in enabling information retrieval across the entire context window, not only for tokens near the beginning of the input sequence.

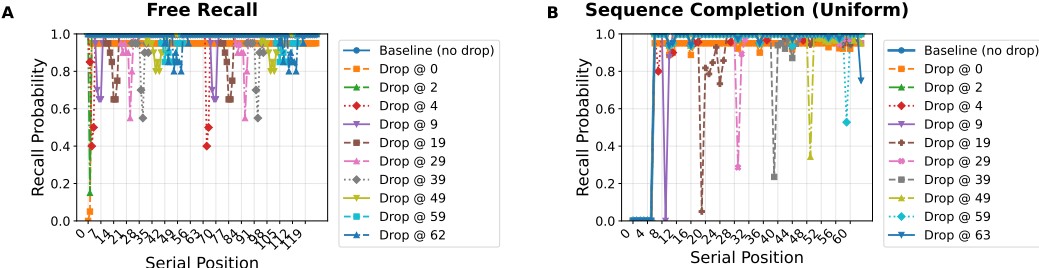

Figure 10: Serial Position Curves with Attention Dropout. (A) Serial position curve for GPT-2 Small evaluated on the free recall task. (B) Serial position curve for GPT-2 Small evaluated on the uniformly-sampled masked sequence completion task. Each curve corresponds to attention disruption at different serial positions throughout the input sequence. We find that attention disruption leads to a local negative impact to recall performance in all cases except position 0 (i.e., the attention sink), which leads to a consistent negative impact across the entire sequence.

