# OpenReview forum: "Lost in the Middle: An Emergent Property from Information Retrieval Demands in LLMs"
_ICLR.cc/2026/Conference — Submitted to ICLR 2026_

### Official Review · Reviewer_Bp4r · 2025-10-18

**Soundness:** 3
**Presentation:** 2
**Contribution:** 3
**Rating:** 6
**Confidence:** 3

**Summary:**

The paper examines the now well know "lost in the effect" of LLMs and show that the phenomenon originates from training (akin to objectives in psychology) that maximizes recency objectives as well as maximizing some long term objectives using an autoregressive model. The paper shows the effect by first constructing the learning from scratch experiment using synthetic data and later on turn to language modelling real use cases.

The paper is able to notably contribute to the understanding of "primacy" effect and shows that 1) the primacy effect is due to autoregressive training with long term tasks 2) the primacy effect is strongly related to the attention sink phenomenon

**Strengths:**

Although the hypothesis has been around in the community for a while for the root case of "lost in the middle effect", this is the first paper that I have seen to illustrate the cause clearly convincingly without involving other causal factors where the paper 1) trains from scratch 2) using synthetic almost toy tasks to illustrate this.

The experimental designs are set up nicely in this paper. The paper investigates into if primacy is related to the autoregressive training and examines a positive case (RNN) and a negative case (T5) for this which are well chosen since the architecture is significantly different to be convincing and informative. The attention sink link is built with its own contribution as well.

**Weaknesses:**

I think the organization of the paper can be made better.

In terms of what I want to see more: in the discussion section, the paper mentions quite some related works mitigate the "lost in the middle effect". The paper posits that intervention on positional attention should have more effect when applied to long term tasks; however, the paper does not state which techniques exactly and don't performs experiments and cite related works for if what the paper posits are true. It would be something informative for the readers.

In terms of what I think can be simplified: the paper examines 3 metrics, however, they seem to connect to each other and lag is not very much used in the texts. Furthermore, although it is valuable to examine real cases (3.4); the conclusion is not expected to differ since the settings are very similar and the phenomenon is relatively well known.

**Questions:**

None

---

> ### Author Response · Authors · 2025-12-04
>
> We appreciate your feedback on our work. We’re encouraged that you viewed our paper as “the first to illustrate the root cause of the lost-in-the-middle effect clearly and convincingly,” found the experiment design “nicely set up,” and considered the model architectures “well chosen.” We address your specific points below.
>
> ---
>
> ### Weaknesses
>
> **Connecting Mitigation Strategies to Our Claims**
>
> Thank you for highlighting this; we’ve clarified this point in our revised discussion. Our claim is that mitigation methods that explicitly reshape positional attention (e.g., RoPE rescaling, attention-offsetting, context-reordering, or position-agnostic training) should matter most when the task demands uniform long-range retrieval, because this is when primacy and sink-based mechanisms play a functional role. While a full sweep of mitigation techniques is beyond the revision scope, we now cite the relevant works and pose the confirmation of these potential effects as a possible future work.
>
> **Metric selection and potential simplification**
>
> Thank you for pointing this out. These metrics serve distinct roles in our analysis and are customarily used in the cognitive science literature that informs our fundamental approach. It is true that these metrics connect to one another, but as evidenced in the change of behavior in PFR, but not SPC, for Llama-3.2 1B (L486 in the Discussion), they each serve as important indicators of the model’s underlying behavior.  For the masked-sequence experiments, we present this experiment as a bridge to the next-token prediction as a way of confirming that these results can extend to more naturalistic tasks learned by general LLMs.

---

### Official Review · Reviewer_L5FX · 2025-10-31

**Soundness:** 3
**Presentation:** 3
**Contribution:** 2
**Rating:** 2
**Confidence:** 4

**Summary:**

This paper investigates the "lost-in-the-middle" phenomenon, where Large Language Models (LLMs) often struggle to recall information from the middle of their context window. The authors posit that this U-shaped recall curve is not an inherent architectural flaw but rather an emergent property from training on mixed retrieval objectives. To test this, they train several models (GPT-2 Small/Large, Llama 3.2-1B, RNNs, and T5) from scratch on synthetic memory tasks. They demonstrate that tasks requiring "free recall" tend to produce a primacy effect (good recall at the beginning), while "running span" tasks produce a recency effect (good recall at the end). When trained on a mixture of these objectives, the models exhibit the full U-shaped curve.

**Strengths:**

**Controlled Experimental Setup:** Training models from scratch on well-defined, synthetic tasks is a significant strength. This methodology provides a clean environment for causal attribution, effectively isolating the training objectives as the primary variable and avoiding the confounding factors of large, diverse pre-training datasets.

**Systematic Ablation Studies:** The paper provides valuable mechanistic insights, particularly through the attention sink dropout experiments. These studies effectively demonstrate how different components (like attention sinks) contribute selectively to different task-driven biases (e.g., primacy).

**Inclusion of Multiple Architectures:** Testing across decoder-only (GPT, Llama), encoder-decoder (T5), and RNN seq2seq models helps to disentangle architectural contributions from training-objective effects, strengthening the paper's claims about the generality of the phenomenon.

**Weaknesses:**

While the paper presents a novel and interesting hypothesis, its conclusions would be strengthened by addressing the following points regarding the generalizability and implications of the findings.

1. Generalizability and Model Scale A key point of tension arises from the Llama 3.2-1B results (Fig 3G, H). These larger, more modern models do not seem to exhibit the same pronounced U-shaped phenomenon observed in the smaller GPT-2 models. This divergence raises crucial questions about the paper's central claim:

Does this suggest the 'lost-in-the-middle' effect, as framed here, is primarily an artifact of limited model capacity rather than a fundamental property of the training objective?

The paper would be significantly strengthened if the authors could elaborate on how their explanation generalizes to modern, more capable architectures that seem to mitigate this issue. If the phenomenon disappears with scale, it suggests the retrieval-objective trade-off may not be the primary bottleneck for today's SOTA models.

2. Situating the Work within Existing Literature The paper would benefit from a more thorough engagement with related literature on positional biases. For instance:

Recent work (e.g., "Transformers need glasses!" [2406.04267], "Why do LLMs attend to the first token?" [2504.02732]) has also explored primacy and recency effects, attributing them to different mechanisms like "information over-squashing."

Connecting to this adjacent work would help clarify the unique contributions of this paper's retrieval-objective hypothesis versus other explanations.

3. Implications for Modern Architectures The reliance on older architectures like GPT-2 (2019) and RNNs, while valuable for the controlled comparison, makes it difficult to ascertain the implications for current SOTA models.

How do the authors reconcile their findings with models like Gemini 1.5 Pro, which demonstrate near-perfect recall across million-token contexts?

A discussion on this point is essential. Are the mechanisms identified here still at play, but simply overcome by sheer scale? Or do modern architectures (e.g., different positional encodings, data mixtures, and attention mechanisms) fundamentally resolve the retrieval-objective trade-off described in this work?

**Questions:**

To help clarify the paper's contributions, I would encourage the authors to elaborate on the following in a revised version:

1. What are the unique contributions of the mixed-retrieval-objective hypothesis when viewed against prior work on positional biases and information "squashing"?

2. What are the practical implications of these findings? Do the results suggest concrete, recommended changes to LLM architectures or pre-training data mixtures?

3. How does the paper's thesis account for the capability of modern SOTA LLMs (and the paper's own Llama 3.2 results) that seem to largely solve the 'lost-in-the-middle' problem at scale? Is the U-shape a temporary phase of training/scale, or a fundamental trade-off that is simply "solved" by other, more dominant mechanisms in larger models?

---

> ### Author Response · Authors · 2025-12-04
>
> Thank you for your constructive feedback and insightful questions. We are glad that you find i) our hypothesis “novel”, ii) our model training methods "effective" and providing a “clean environment for causal attribution”, iii) the insights regarding the attention sinks “valuable”, and (iv) our results “generalizable” across architectures. Below we will address your questions in detail.
>
> ---
>
> ### Weaknesses
>
> **Generalizability and Model Scale**
> Please refer to Q1. in the General Response.
>
> We have expanded the paper to clarify what the Llama-3.2–1B results imply (L486). Although the 1B model has a flatter SPC, its recall strategy still shifts under different retrieval demands (e.g., its PFR curve changes under the combined task). This indicates that underlying mechanisms remain active, despite optimal recall. We have also added experiments across model scales and longer context lengths. These show a reduction of the U-shape with scale consistent with prior work reporting reduced lost-in-the-middle in larger LLMs (appendix section A.2). This suggests the reduction is not due to the absence of the mechanism, but to larger models compensating for positional biases.
>
> **Explanations for modern architectures**
> Please refer to Q3. in the General Response.
>
> Our expanded scale sweep shows that larger models exhibit reduced U-shape but still display the same behavioral shifts under changing retrieval demand (e.g., PFR changes, sink sensitivity). Model scale reduces symptoms, not the underlying interaction between retrieval demands and causal processing (Appendix A.2). We have also expanded the Discussion to include more recent papers (L465). Our results complement these more theoretical works by showing when these architectural tendencies have functional behavioral impact under specific retrieval objectives. One of our central findings is that attention sinks support long-range, but not short-range, retrieval tasks.
>
> **Implications for modern architectures**
> In the revision, we added explicit discussion connecting our findings to other recent explanations (L476). Our results show when these biases matter depends on the training objective. Regarding modern systems, we add discussion that high-capacity LLMs can mask positional biases even while the underlying forces remain. Our scale sweep shows that primacy/recency decreases steadily with increasing capacity (appendix A.2). Literature reports that architectural changes, new positional encodings, and diverse training mixtures further mitigate positional bias. These observations suggest that the mechanisms we identify still exist, but their influence is outweighed by scale and design choices in large models. Our minimal tasks clarify why lost-in-the-middle arises and why SOTA systems often do not express it.
>
> ---
>
> ### Questions
> **Unique contributions of the mixed-retrieval-objective hypothesis**
> Our hypothesis contributes by showing that architectural biases alone do not determine whether a U-shape emerges. Instead, the training objective determines which biases become behaviorally expressed. Our results explain *when* that early-token influence matters: primacy arises under long-term retrieval demand, recency under short-term demand, and the full U-shape under mixed demands. This framing clarifies why positional effects vary across tasks, architectures, and scales, and provides a basis for predicting mitigation effectiveness. We have added a dedicated discussion addressing this (L466).
>
> **Practical implications**
> Our results suggest two practical implications. First, the retrieval demands in the pre-training mixture matter: shifting the balance between long-range and short-range retrieval changes whether primacy, recency, or neither becomes dominant in task performance. This provides a means to anticipate when positional-bias mitigation techniques will help. Second, the analysis highlights which components primarily drive long-range retrieval bias (causal-style attention and sink formation) indicating that positional encodings, attention routing, and model scale can be tuned to reduce reliance on early-token anchors. While we believe that our current results suggest these potential changes, we have added a mention of experiments confirming this to Future Work.
>
> **SOTA performance**
> Our view is that the U-shape reflects an interaction between retrieval demands and causal processing, but one that can be overridden by capacity, positional-encoding improvements, and diverse training mixtures. Our Llama-3.2 results illustrate this: the serial-position curve flattens, but the model’s retrieval strategy (e.g., PFR behavior, sink sensitivity) still responds to differing demands. This indicates the mechanism continues to operate, but its behavioral impact is reduced with scale; Modern SOTA systems push this further through scale and architectural refinement. We have added this explanation to the Discussion section.

---

### Official Review · Reviewer_Xfd2 · 2025-11-01

**Soundness:** 3
**Presentation:** 3
**Contribution:** 2
**Rating:** 4
**Confidence:** 4

**Summary:**

The paper draws parallels with resource-rational perspectives in cognitive psychology, explaining the recency and primacy effect usually observed in autoregressive LLMs. It interprets the positional bias as an emergent behavior arising due to different memory demands and architectural design.

It trains model from scratch with minimal training tasks to show emergence of U-shaped curve. Their analysis reveals that training for free recall task shows primacy effect and short-term running span task shows recency effect. And combining both tasks yield U-shaped curve, also referred to as lost-in-the-middle effect.

The paper performs controlled study to show these effects over 3 small to medium scale autoregressive LLMs. Where the U-shaped phenomena is weakened for the largest model (LLama 3B). This is attributed to the large scale of the model. The paper shows that causal architecture is one of the main reason for the emergence of lost-in-the-middle effect and it disappears when using bidirectional encoder-decoder architecture. The causal architecture leads to the formation of attention sinks which is responsible for the primacy effect. The paper performs ablation on attention sinks by destroying stronger sinks to confirm their effect.

**Strengths:**

1. The work proposed an isolated setup with minimal tasks to demonstrate the formation attention sinks using a causal architecture. This is a novel contribution of this work. The memory demand task design is simple and easy to interpret.

2. The study makes clear observations by training the model from scratch. It makes a convincing case for the role of short- and long-term memory demands and for the role of causal model for the observed primacy and recency bias.

3. The study conducts thorough empirical analysis with different memory demands. The study disrupting attention sinks with long-term memory demands is particularly insightful. Additional experiment with masking strategy shows that the observations are generalizable.

4. The paper is well-structured and easy to read.

**Weaknesses:**

1. Large-scale experiment is only limited to one Llama model which does show weak U-shaped curve. It is unclear whether the weakened U-shape performance is due to the scale of the model, model architecture or training dynamics. More baseline at multiple scales are needed to confirm the effect of scaling.

2. While the proposed memory tasks allow clean analysis, but it is a much simplified task compared to real LLM tasks. Although masked sequence completion does show similar U-shaped behavior, it is unclear whether such attention-sink issue arise in general-purposed LLMs in practice. For example, do recent large-scale (>10B) general-purpose LLMs trained with standard next-token prediction format also show this U-shaped effect?

3. The contribution of this work are unclear. Which experiment setup and evaluation metrics are novel in this work? Please clarify which findings are confirmations of previous work and which are new insights (include comparison to Liu et al 2023 and Wu et al. 2025) . Also, is the usage SPC, PFR, and CRP to study positional bias novel to this work or has it been proposed before?

Some more questions are included in the Questions section below.

**Questions:**

1. It is an interesting observation that larger models show flattened U-shaped recall curves. It is unclear how scaling changes the attention pattern and distributes it more evenly. A quantitative analysis of attention distribution and sink strength would clarify this observation.

2. Wu et al. 2025 already showed in their work that causal nature of the autoregressive model is responsible for primacy effect due to attention sink formation. It is unclear what new insights does this study offer in this regard specifically?

3. It is insightful to know that attention sinks formed at early positions in causal models are responsible for the primacy effect. However, it is unclear how do the formation of these attention sinks occur and why only in the early tokens.

---

> ### Author Response · Authors · 2025-12-04
>
> Thank you for your constructive feedback and insightful questions.
>
> ---
>
> ### Weaknesses
>
> **Large-scale experiment is only limited to one Llama model**
>
> Please refer to Q1. [Generalizability and model scale] in General Response. We agree that isolating the source of the weakened U-shape requires looking across multiple scales. In the revision, we have added experiments with additional model sizes and included a plot showing how U-shape strength varies with scale. These results indicate that the attenuation is primarily scale-related rather than specific to Llama architecture or training dynamics, and they align with prior observations that larger models exhibit reduced positional bias (see appendix section A.2).
>
> **Simplified tasks versus natural LLM tasks**
>
> Our goal here was to isolate the minimal task conditions under which the U-shape effect emerges. The U-shaped effect has already been observed in large general-purpose LLMs trained with standard next-token prediction (in addition to the appearance of attention sinks), so our aim was to explain why it reliably appears rather than re-demonstrate it. Our masked-sequence experiments serve as a bridge toward natural pretraining, showing that when next-token prediction is paired with the mixed retrieval demands, the same effects emerge. We have added references to related works that have previously investigated the U-shaped phenomenon and attention sinks (but not necessarily the link between them), which have shown the U-shaped effect appearing in larger, general-purpose LLMs (L484 in the Discussion section).
>
> **Clarification of contributions and novelty**
>
> We’ve revised the Discussion section to explicitly separate what is confirmatory from what is new and to compare against Liu et al. (2023) and Wu et al. (2025) (L466)). Liu et al. document the U-shape in natural long-context tasks, and Wu et al. analyze how causal masking leads to positional bias; our main new contributions are: (i) a minimal task framework (free recall, running span, and combined task) that shows how specific information-retrieval demands generate primacy, recency, and the full U-shape, (ii) a masked sequence-completion task that connects these retrieval demands to a more standard next token prediction task, and (iii) causal evidence that attention sinks functionally support primacy and long-range recall by sink ablation studies (plus the architecture comparison (autoregressive RNN vs T5) under identical tasks). SPC-, PFR, and CRP-style curves have been used extensively to study these effects in cognitive science experiments, but their use in analyzing this effect in LLMs is relatively recent.
>
> ---
>
> ### Questions
>
> **Scaling and flattened U-shaped curves**
>
> Please refer to Q1. [Generalizability and model scale] in the General Response.
> Our results (appendix A.2) show flatter serial-position curves in larger models. Without deeper attention-level analysis we cannot determine exactly how attention allocation changes with scale. Prior work suggests that scale smooths positional biases and reduces early-token anchoring. A systematic comparison of sink magnitude and attention dispersion across layers and scales would directly test this, and we added it to Future Work.
>
> **Relationship to Wu et al. (2025)**
>
> Please refer to Q3. [Novelty compared to existing work] in the General Response.
> Wu et al. provide a mechanistic account linking causal masking to positional bias. Our work builds on that by showing when these architectural biases actually produce primacy. We show that primacy emerges only when causal architectures are paired with uniform long-term retrieval demands, and disappears in autoregressive models trained on short-term retrieval tasks. We also demonstrate that primacy, recency, and full U-shapes can be induced or removed by manipulating task demands alone, and that attention sinks selectively support long-range retrieval via ablation studies. More clarification is added in the Discussion (L476).
>
> **Formation of attention sinks and early-token localization**
>
> Prior work (e.g., Wu et al. 2025; Gu et al. 2024) shows that sinks arise because causal masking and residual-stream accumulation make early tokens persistent anchors that later states can always attend to. Our contribution is to show the functional impact. When tasks place uniform long-term retrieval demands on the sequence, the model uses these early anchors to support primacy, whereas tasks emphasizing recent information do not rely on sinks. A full analysis of why sinks localize at early positions remains open, but our results help clarify when and why these structures become behaviorally relevant.

---

### Official Review · Reviewer_Ni46 · 2025-11-02

**Soundness:** 3
**Presentation:** 3
**Contribution:** 3
**Rating:** 6
**Confidence:** 3

**Summary:**

The paper studies the “lost-in-the-middle” effect in LLMs and argues it is an emergent adaptation to information-retrieval (IR) demands during training rather than merely a flaw. The authors train GPT-2 (small/large) and Llama-3.2-1B from scratch on toy but principled human-memory paradigms: Free Recall (uniform long-term demand) and Running Span (end-weighted short-term demand). Training on both yields a U-shaped serial-position curve (primacy + recency), i.e., lost-in-the-middle (Figure 2C; Figure 3G–I). They further: (i) link primacy to autoregressive processing by showing strong primacy in an RNN seq2seq but not in a bidirectional T5 (Figure 4), and (ii) implicate attention sinks by ablating heads/layers flagged by a sink metric, which selectively degrades tasks with long-term retrieval demand and removes primacy (Figure 5D–G). Finally, they replicate the phenomena in a masked sequence completion task that more closely resembles next-token prediction (Figure 6). Formal task definitions and additional ablations are in the Appendix.

**Strengths:**

Originality: Elegant, minimal task-design lens that unifies primacy/recency and lost-in-the-middle; neat architectural contrast (RNN/T5) and sink-based analysis. (Figures 2–6).
Quality: Multiple models (GPT-2 S/L, Llama-3.2-1B) trained from scratch; consistent behavioral metrics; ablations that selectively hit long-term retrieval tasks (Figure 5G; Figure 6G).
Clarity: Clear exposition; formal task definitions; helpful visuals of sink heatmaps and SPC/PFR/CRP curves (pp. 6–8, 12–13).
Significance: Offers a theoretical framing likely to influence evaluation/mitigation strategies and spur work that tailors training curricula or architectures to desired IR profiles.

**Weaknesses:**

External validity: No natural-data or real long-context benchmarks (QA/book-sum/code) to demonstrate that the mechanism quantitatively explains or predicts behavior in practice.
Ablation method: “Attention-sink dropout” removes entire attention layers flagged as sinks; this may simultaneously remove useful computation. More surgical interventions (head-level masking, key-positional rescaling) or counterfactual re-routing would strengthen causal claims.
Sensitivity & controls: Limited analysis of ε thresholding (0.8), dependence on context length, tokenization, or positional encodings (e.g., RoPE vs absolute). The combined-task loss weighting (FR vs RS) and its effect on the U-shape are not detailed.
Scale: Only up to ~1B parameters; the paper notes reduced U-shape at larger scale—directly charting trend vs scale/length would be valuable.

**Questions:**

Generalization: Can you test the IR-demand hypothesis on natural long-context suites (e.g., re-ordering/context-insertion diagnostics) to predict when mitigations help?
Loss mixing: How are FR and RS combined—same batch, same loss weight? What happens when you sweep the mixing ratio; does the U-shape vary smoothly?
Sequence length: Does primacy’s dependence on attention sinks persist at much longer contexts (e.g., 4k–32k)? Any qualitative shifts?
Sink metric robustness: Results vs different ε thresholds and identification methods? What about ablating non-sink heads/layers matched for magnitude to isolate the sink-specific effect?
Positional encodings/architectures: How do RoPE variants, ALiBi, or state-space and hybrid encoder-decoder models affect primacy/recency under the same tasks?
Surgical ablations: Could you replace sink layers with frozen copies or attention reweighting to rule out capacity loss as the driver?
Data realism: If you train on skewed temporal distributions (e.g., recency-heavy Zipfian windows approximating web/news), does the learned curve match your predicted balance of primacy/recency?

---

> ### Author Response · Authors · 2025-12-03
>
> Thank you for your constructive feedback and insightful questions.  Below we address your questions in detail.
>
> ---
>
> ### Weaknesses:
>
> **External validity:**
> We agree that validating mechanisms on natural long-context benchmarks is important, but the lost-in-the-middle effect has already been documented on several real datasets, and our goal was to isolate the minimal task conditions under which the phenomenon emerges. By stripping away confounding factors present in naturalistic tasks, we aim to pinpoint which retrieval demands and architectural features are causally responsible for the effect. We also bridge the gap between controlled recall experiments and natural settings using the masked sequence-completion task, which preserves next-token prediction structure while allowing controlled manipulation of retrieval demands. We acknowledge this point of concern and have highlighted that prior work on the U-shaped effect has hinted at its presence in more naturalistic datasets (L265). We pose direct confirmation of retrieval demands producing this effect in natural data as a direction for future work.
>
> **Ablation method:**
> Please refer to Q2. [Alternative attention sink ablation experiments] in the General Response.
>
> In response to your concerns, we implemented two more targeted ablations (per-head masking of identified sink heads and key–positional rescaling) following your suggestions. Both interventions reproduce similar patterns to our original findings: primacy and long-term retrieval degrade selectively, while short-term retrieval remains largely unaffected. These new analyses have been added to appendix section A.3.
>
> **Sensitivity and controls:**
> We have clarified why we chose a single value for ε thresholding. A threshold of 0.8 cleanly isolates layers containing clear attention sinks, while lower thresholds broaden ablation to many more layers and cause nonspecific degradation. We clarify that FR and RS losses were weighted equally during combined-task training (added to Task Definitions). Although a full sweep over tokenization, positional encodings, and context lengths is beyond scope, we now report results from additional model families with differing encodings and longer training contexts. These models show similar primacy/recency/U-shape patterns (section A.2), providing evidence that these effects generalize beyond the presented architectures.
>
> **Scale:**
> Please refer to Q1. [Generalizability and model scale] in the General Response. In revision, we included additional models spanning a wider parameter range and added a plot quantifying U-shape strength as a function of scale (Figure 8, appendix A.2). The trend aligns with our and prior observations: primacy and recency weaken with increased scale, while overall recall improves.
>
> ---
>
> ### Questions:
>
> **Generalization:**
> Since natural long-context benchmarks already show the U-shape (and multiple works provide mitigation strategies), our focus was identifying minimal conditions that reproduce the effect to expose underlying causes. We include direct predictive tests on natural suites as future work.
>
> **Loss mixing:**
> We added clarification that FR and RS were combined with equal loss weighting in shared batches (Task Definitions). While a full sweep is outside the revision scope, our IR-demand hypothesis suggests a smooth trade-off between primacy and recency under different mixing ratios. This would be a natural extension for future work.
>
> **Sequence length:**
> Our new scale experiments (section A.2) modestly extend context (64→256 items), and within this range the primacy–sink relationship remains stable. Because attention sinks and lost-in-the-middle behavior both scale with sequence length in prior work, evaluating the interaction at 4k+ contexts would clarify whether sink-driven primacy strengthens, saturates, or changes qualitatively. We mention this in the Future Work section.
>
> **Sink metric robustness:**
> We clarify the use of ε=0.8. As shown in Fig. 5A–C, ε=0.8 is the only value that cleanly isolates genuine sink heads. Lower values add many more layers to the ablation, causing broad degradation that obscures sink-specific effects.
>
> **Positional encodings and architectures:**
> Our revision includes results from additional model families with varying scales and positional embeddings, and the primacy/recency patterns persist (appendix A.2). A broader sweep over other architectural components is beyond scope, but we list these as directions for future work.
>
> **Surgical ablations:**
> We added more targeted ablations (head masking and key–positional rescaling), which reproduce the main effects (appendix A.3). More extensive study of attention sink layers is a promising future direction.
>
> **Data realism:**
> Our masked-sequence experiments show that sampling distributions shift the primacy/recency balance, and extending this to more realistic temporal statistics is a natural follow-up. We mention this in Future Work.

---

### Author Response · Authors · 2025-12-03
**General Reviewer Response**

We thank all the reviewers for their valuable and constructive comments. Lost-in-the-middle is an important phenomenon to understand in LLMs. We are glad that the reviewers find (i) our approach “novel” and “effective” (Ni46, Xfd2, L5FX31, Bp4r), (ii) our results “convincing” in demonstrating the “causal contribution” and “root cause” of information-retrieval demands (Xfd2, L5FX31, Bp4r), (iii) our findings “generalizable” and tested over multiple model architectures (Ni46, Xfd2, L5FX31), and (iv) our paper “well-written” with “clear exposition” (Ni46, Xfd2).

Before addressing each reviewer’s comments in detail, we highlight three common comments identified across reviewers and our responses.

---

### Q1. Model scale (Ni46, Xfd2, L5FX)

To assess the role of model scale, we trained several larger open-weight models from scratch on the same task suite used in the main paper: Gemma-2 2B, Qwen-2.5 1.5B, and Llama-3.2 3B, and added these results to Appendix A.2. These models span distinct architectural design choices and positional-encoding variants.

Consistent with our previous conclusion based on GPT-2 small, GPT-2 large, and Llama 2-1.3B, the lost-in-the-middle effect also emerges from these additional models. The degree of the effect reduces with increasing model scale. We quantify this trend using a U-shape index, which declines steadily from GPT-2 to Gemma-2, Qwen-2.5, and Llama-3.2, demonstrating that the U-shape pattern decreases with increasing model complexity. These results indicate that the attenuation is primarily scale-related rather than specific to the Llama architecture or training dynamics, aligning with prior observations that larger models exhibit reduced positional bias (see Appendix A.2).

---

### Q2. Validation over more naturalistic tasks (Ni46, Xfd2)

We used well-defined and synthetic tasks to train models from scratch. Two reviewers suggested using more naturalistic tasks to demonstrate the proposed mechanisms. We argue that using simplified rather than naturalistic tasks is a strength rather than a weakness of the present work:

* Previous work has already demonstrated the existence of the lost-in-the-middle effect in more naturalistic tasks. The goal of the current work is to isolate the minimal task conditions under which the lost-in-the-middle phenomenon emerges. By stripping away the confounding factors present in naturalistic tasks, we’re aiming to pinpoint which retrieval demands and architectural features are causally responsible for the effect.

* To bridge the gap between controlled memory experiments and more natural tasks,  we replicated our results using a masked sequence completion task, which more closely resembles the next-token prediction process in real-world LLM pre-training.

While two reviewers (Ni46, Xfd2) viewed the simplicity of our experimental design as a potential weakness, the other two reviewers highlighted this same design's contribution:

* **L5FX:** “ Training models from scratch on well-defined, synthetic tasks is a significant strength. This methodology provides a clean environment for causal attribution, effectively isolating the training objectives as the primary variable and avoiding the confounding factors of large, diverse pre-training datasets. ”
* **Bp4r:** “This is the first paper that I have seen to illustrate the cause clearly convincingly without involving other causal factors where the paper 1) trains from scratch 2) using synthetic almost toy tasks to illustrate this.”

---

### Q3. Novelty compared to existing work (Xfd2, L5FX)

Two reviewers requested clarification on which findings are confirmations of prior work and which are new insights.

* We’ve revised the Discussion section to explicitly separate which findings are confirmatory from what are new. Liu et al. document the U-shape in natural long-context tasks, and Wu et al. analyze how causal masking leads to positional bias; our novel contribution is to identify  a minimal task framework (free recall, running span, and combined task) that shows how specific information-retrieval demands generate the lost-in-the-middle effect.

* We added a new paragraph in the Related Work section to directly compare our work with Wu et al. (2025) and Barbero et al. (2024). These mechanistic accounts describe why positional bias emerges from architectural constraints. Our results distinguish from these perspectives by identifying when these biases affect downstream task performance and behavior. This helps reconcile why positional biases are visible in some settings and attenuated in others, even within the same architecture.

* The way we draw our key hypothesis based on cognitive science literature is novel. While prior work has noted the similarity of the lost-in-the-middle effect in LLMs and humans, our paper is the first in transferring what we know about human memory, i.e., what contributes to the lost-in-the-middle effect in humans, to systematically test these hypotheses in LLMs.

---

### Meta-Review · Area_Chair_VjhG · 2026-01-07

**Summary:**

The paper proposes that the "Lost-in-the-Middle" (LitM) phenomenon in LLMs is an emergent adaptation to conflicting information retrieval demands during pre-training. By training models (GPT-2, Llama variants) from scratch on synthetic cognitive science paradigms, Free Recall (long-term demand) and Running Span (short-term demand), the authors demonstrate that LitM emerges when these demands are combined. They identify two key mechanisms: Recency aligns with short-term retrieval demands, while Primacy emerges from long-term demands interacting with autoregressive architecture and attention sinks. The submission further validates these findings using a masked sequence completion task and ablation studies.

The primary concerns identified across the reviews centered on model scale and practical significance. While the experimental design was praised for its "clean" isolation of variables, high-confidence reviewers (L5FX, Xfd2) questioned whether LitM is a fundamental property of the architecture or merely an artifact of limited capacity in smaller models. Specifically, the concern was that if the phenomenon disappears in state-of-the-art models (as suggested by prior literature), the mechanism described here might be of limited relevance to the current frontier of foundation models. Reviewers also noted the reliance on synthetic tasks, questioning the transfer of these findings to naturalistic pre-training distributions.

Based on these concerns and the rebuttal data, I recommend *Rejection*. The authors provided a robust rebuttal that included training larger models (up to Llama 3.2 3B). However, this new data confirmed the trend identified by the reviewers: the "U-shape index" steadily declines as model size increases. While the authors argue the underlying mechanism persists even when behavioral symptoms diminish (being compensated by capacity), this interpretation does not adequately address the practical relevance concern raised by the negative reviewers.

This decision balances methodological rigor against significance. The paper is scientifically sound and the experimental design is elegant. However, the high-confidence reviewers (L5FX, Xfd2) correctly identified that the central phenomenon is transient, and it disappears as models scale up. While mechanistic understanding has intrinsic value, the paper does not demonstrate that these insights lead to actionable improvements for larger models or predict when mitigations will be effective. Consequently, the contribution is primarily of theoretical interest regarding small-model dynamics rather than current impact for the foundation model community.

**Reviewer Concerns:**

### Addressed concerns:
* The reviewer Ni46 noted that dropping entire layers was too coarse. The authors added Appendix A.3 with "surgical" ablations (head masking/key rescaling), which satisfied this technical critique.
* The reviewer Xfd2 raised novelty concerns. The distinction from Wu et al. (2025) was clarified in the revision, distinguishing the mechanistic cause (causal masking) from the functional emergence (retrieval demands).

### Outstanding Concerns:
* Reviewers L5FX and Xfd2 raised concerns about the significance and the scale of the reported results, which remains the pivotal issue here. The rebuttal added Appendix A.2, showing the U-shape index drops significantly with model scale. While the authors argue the mechanism is still detectable, the problem effectively vanishes. This strengthens the negative reviewers' position that the work lacks urgency or impact for the current generation of models.
* Reviewers Ni46 and Xfd2 argued against validity, since the bridge between synthetic memory tasks and real-world "wild" pre-training remains theoretical. The masked sequence completion task is a good proxy, but does not fully resolve whether these mechanisms dictate behavior in complex natural language tasks.

**Reviewer Scores:**

* **Reviewer Ni46. Original score: 6. Predicted score: 6**. The authors addressed the specific technical critique regarding ablation. However, the reviewer's broader concerns about external validity and naturalistic benchmarks remain outstanding, making a jump to the next tier (8) unlikely.
* **Reviewer Xfd2. Original score: 4. Predicted score: 4**. The novelty was clarified, but the scale experiments confirmed their doubts about relevance.
* **Reviewer L5FX. Original score: 2. Predicted score: 2**. This reviewer’s primary critique was that LitM is an artifact of scale. The rebuttal provided data (Appendix A.2) that supported this view. They are unlikely to increase their score.
* **Reviewer Bp4r. Original score: 6. Predicted score: 6**. They were positive but had lower confidence; likely remains unchanged.

---

### Decision · Program_Chairs · 2026-01-26

Reject